# A Model of Motivational and Technological Factors Influencing Massive Open Online Courses' Continuous Intention to Use

**Samer Ali Al-shami [1,\*], Salem Aldahmani [1], Massila Kamalrudin [2], Nabil Hasan Al-Kumaim [3], Abdullah Al Mamun [4], Mohammed Al-shami [5] and Mustafa Musa Jaber [6,7]**

[1] Institute of Technology Management and Entrepreneurship, Universiti Teknikal Malaysia Melaka, Melaka 76100, Malaysia; p081810036@student.utem.edu.my

[2] Fakulti Teknologi Maklumat dan Komunikasi, Universiti Teknikal Malaysia Melaka, Melaka 76100, Malaysia; massila@utem.edu.my

[3] Faculty of Technology Management and Technopreneurship, Universiti Teknikal Malaysia Melaka, Melaka 76100, Malaysia; nabil@utem.edu.my

[4] Graduate School of Business, Universiti Kebangsaan Malaysia, Bangi 43600, Malaysia; almamun@ukm.edu.my

[5] Yemen Faculty of Education, Sana'a University, Sana'a 1247, Yemen; mohammedalshami075@gmail.com

[6] Department of Medical Instruments Engineering Techniques, Al-Turath University College, Baghdad 10021, Iraq; mustafa.musa@duc.edu.iq

[7] Department of Medical Instruments Engineering Techniques, Alfarahidi University, Baghdad 10021, Iraq

\* Correspondence: samshami79@gmail.com

**Abstract:** Massive open online courses have been regarded as effective technological innovations that improve educational systems in the era of digitalisation. However, only 10% of the registered students complete their courses. This study aims to examine the motivational and technological factors and contextual features on students' continuous intention to use. A questionnaire was gathered from 315 of students in the UAE and revealed that social motivational and technological factors driven by the technology acceptance model and technology task fit theory significantly influenced the students' continuance intention to use. This study also revealed that contextual features including language use and course accreditation are important indicators determining students' behaviours toward the use. Hence, this study proposed an integrative model to explain ways to improve continuance intention to use. This study contributes to the sustainable use of massive open online courses in developing countries through an integrative model.

**Keywords:** motivational; technological; factors; massive; open; online; courses; intention

## 1. Introduction

Massive open online courses (MOOCs) are an effective innovative technology [1,2], that drives the development of the education sector [3]. The underlying logic is that MOOCs easily bridge the educational gap between students regardless of their socioeconomic status by offering open education with minimal or no fees. However, only 10% of enrolled students have completed their MOOC courses [4]. Without sustained, continuous intention to use MOOCs, it is difficult for universities to solicit users' feedbacks to improve the technology. Additionally, it is less likely for the developers and service providers to attain financial benefits, such as revenue from advertisements, in-app purchases, subscriptions, and sponsorship [5]. Based on the users' perspective, a lack of continuance intention to use MOOCs may negatively affect education-related behaviours [6]. Zhou [7] incorporated the theory of planned behaviour (TPB) and self-determination theory (SDT) to investigate the factors that might affect students' decisions in using MOOCs. Daneji et

al. [8] employed the expectation–confirmation model (ECM) and the technology acceptance model to examine the impacts of students' perceived usefulness, confirmation, and satisfaction regarding MOOC continuance intention. Meanwhile, Khan et al. [9] employed the task–technology fit model, social motivation, and self-determination theory to determine students' adoption of MOOCs. Lung [10] used the merging theory of planned behaviour and self-regulated learning models to assess students' behavioural intention to use MOOCs. Only limited studies have investigated the factors influencing MOOC continuation intentions, particularly in the Middle East [11].

The United Arab Emirates (UAE), which is the selected case of this research, is one of the most active technological and educational countries in the Arab region. The UAE has a population of 9.77 million and is one of the richest countries in the world based on gross domestic product (GDP) per capita. Regarding technology use, 91% of the residents use mobile Internet and over 98% of the households have Internet access [12].The education sector is one of the fastest growing in the region. The development of smart learning and technology was part of the 2021 Vision agenda to harness human capital to build "a knowledge-based society" [13]. To meet the goals of UAE Vision 2021, the country is reinforcing education primarily based on technology, and hence, is strengthening MOOCs across the region to improve universities' sustainability and students' learning capabilities [14]. In addition, as a result of the COVID-19 pandemic, the Gulf Arab Countries including the UAE enforced the online learning mode across the whole educational system starting in March 2020 [14]. This new embrace of online education is expected to significantly boost the MOOC market in the UAE [15]. Today, many universities, and organisations in the private and public sectors deliver formal and nonformal education through MOOCs [16]. Despite the importance of MOOCs in the development of education system, the students' continuous intention to use is the main axis of its success. This, in turn, leaves a research gap that needs to be addressed in Arab region countries since the majority of studies of MOOCs were conducted in Western countries and only a few in Asian countries.

To develop users' continuance intention to use, users' behaviours to adopt and MOOCs resources should be addressed. The first perspective focuses on the users' attitudes, adoption, and behaviours toward MOOCs, while the latter focuses on how well MOOCs fulfil users' requirements through MOOCs' utility. The two perspectives are linked because the level of utility cannot be accomplished without MOOCs' acceptance. Despite that, the current studies that seize students' willingness to join MOOCs are too crude to integrate these two perspectives [17]. Moreover, MOOCs' features vary between developers and between one study and another [18]. The other gap is associated with users' behaviour to adopt and use MOOCs [19]. In addition, some educational institutions in several countries, such as in the UAE, have customised MOOCs' platforms by offering alternative languages and cost advantages [11]. Despite the importance of contextual factors, limited studies have investigated the effects of contextual factors on students' continuous intention in using MOOCs. Therefore, the purpose of this study is to identify the motivational, technological, and contextual factors and examine their effects on students' continuous intention to use MOOCs in the UAE. First, this study aims to examine the relationship between MOOCs' technological factors (technology task fit and technology individual fit) on students' perceived usefulness and ease of use. Second, this study aims to examine the relationship between social motivational as well as MOOCs' contextual factors and students' perceived usefulness and ease of use. Finally, this study aims to examine the effect of students' perceived usefulness and ease of use on students' attitude toward MOOCs and continuance intention to use MOOCs.

Theoretically, this study provides an integrative model that influence continuous intention to use MOOCs driven by the technology acceptance model (TAM) and task technology fit (TTF). This study extends the literature on MOOCs from the perspective of developing countries, generally, and the United Arab Emirates, particularly, through identifying the contextual factors and their effect on students' continuous intention to use

MOOCs. Finally, this study provides an empirical model that guides MOOCs' developers as well as providers on how to improve education quality and sustainability through improving students' continuous intention to use MOOCs.

## 2. Literature Review

### 2.1. Theoretical Foundation

To date, TAM is used to explain the relationship between motivations toward behaviour and attitude in using technology. TAM was established by Davis to elaborate the influencing factors of technology use toward perceived ease of use and perceived usefulness of a particular technology, which concurrently and jointly influence users' behavioural intention. Despite TAM's advantages in forecasting IS's use in the initial stages [20], [21] and predicting the technology design problems before deployment, it has several imitations. The variance in behavioural intention was explained by only 40% [22], which is considered a partial explanatory influence [23,24]. Thus, the incorporation of external variables into TAM can improve the explanatory influence of this model. Thirdly, it was argued that there is no consistency in the relationships between TAM drivers in varying settings and contexts [23,24]. For instance, some studies discovered a significant correlation between perceived ease of use, perceived usefulness, and behavioural intention, while others found the opposite [24,25]. These studies left an open gap in testing TAM in different contexts using varying respondents. Therefore, Wu and Chen [17] suggested the integration between TAM and TTF to overcome the limitations of each theory. The TTF model focuses on whether the technology is suitable for task completion by examining the adoption of innovative technologies in supporting task requirements' fulfilment [17,26]. Meanwhile, Zhou et al. [27] argued that users are more concerned about whether the technology can satisfactorily perform a task rather than how advanced it is. Thus, the model advocates that a user should only accept a technology if it helps the user complete a task. Even though the TTF model emphasises the technology's capacity to fit the required tasks [28], the available literature has extensively assessed the technology features more compared to tasks' characteristics [29,30].

### 2.2. Hypotheses

Driven by the relationship between TTF, MOOCs' features, social motives, and TAM ease of use and usefulness, we proposed an integrative model that explains students' continuance intention to use MOOCs from the perspective of the Arab region, as shown in Figure 1. The underlying logic is that MOOCs' technological factors, MOOCs' context features, and social motivation influence students' perceived ease of use and usefulness, which jointly influence their attitude toward continuance intention to use MOOCs.

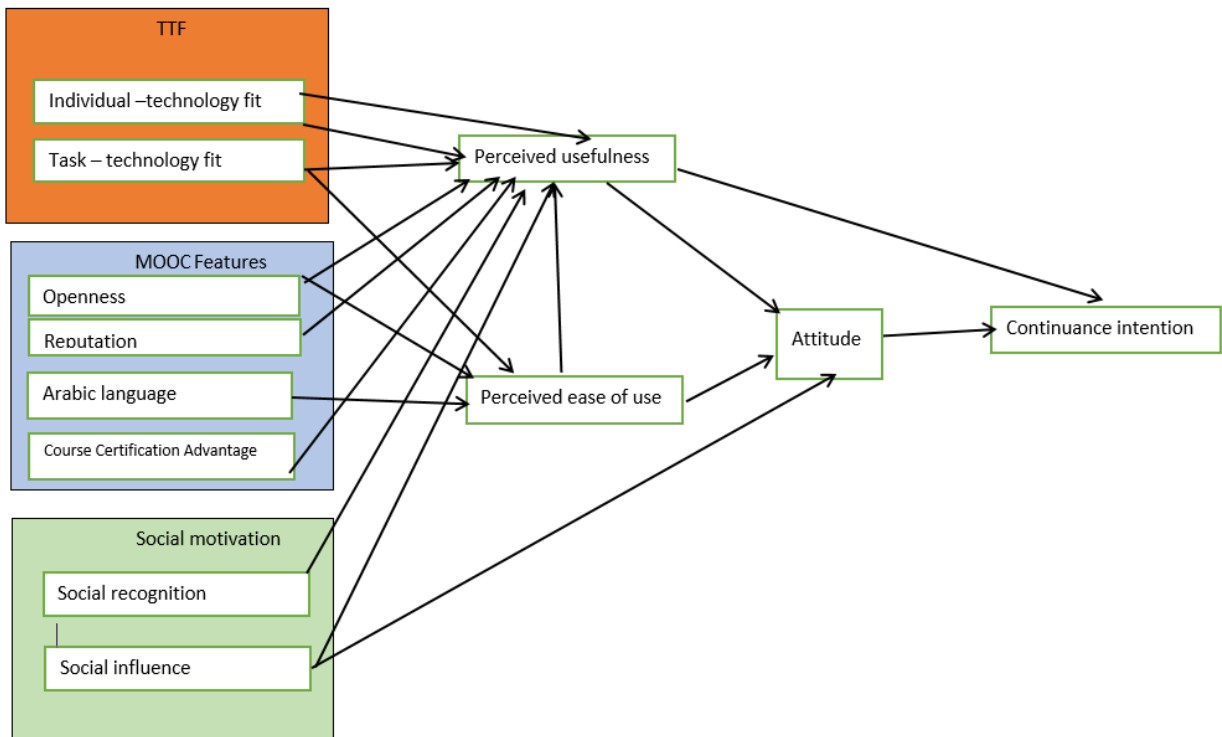

**Figure 1.** Hypothesised and tested associations in this study. Source: Authors.

*2.3. Perceived Ease of Use*

In MOOCs, perceived ease of use refers to a user's belief that MOOCs will be free of burden. The facility to acquire skills through MOOCs is an example of perceived ease of use. Joo et al. [31] revealed that the perceived ease of MOOCs among students impacted the perceived utility. The authors suggested that learner-friendly features can prevent technical problems during the preliminary use of MOOCs. MOOCs are useful for learning activities and purposes. In a similar vein, Ma and Lee [32] revealed that both adopters and nonadopters have a similar impact on their intention to adopt MOOCs (such as perceived usability, value for money, and self-regulation). Therefore, this study hypothesises that:

**Hypothesis 1 (H1).** *The perceived ease of use of MOOCs has a significant positive relationship with the students' perceived usefulness.*

Since MOOCs are easy to use and leave a good impression on the platform and its contents among students, it influences their intention to adopt MOOCs implicitly or explicitly over perceived usefulness. Salloum et al. [33] added that user attitude toward E-learning is influenced by perceived ease of use. Moreover, technology familiarity can sociologically impact users by influencing their attitude to want to use it more. However, Wu and Chen [17] indicated no positive significance between perceived ease of use and attitude to use MOOCs. They stated that web accessibility comprises similar capabilities and features, making MOOCs easy to use. Thus, the student's attitude toward MOOCs' adoption depends completely on its perceived usefulness. This study, however, perceives perceived ease of use as an essential predictor of students' attitude in using MOOCs. Thus, this study hypothesises that:

**Hypothesis 2 (H2).** *The perceived ease of use of MOOCs has a significant positive relationship with students' attitudes.*

*2.4. Perceived Usefulness*

The significance of perceived usefulness is well-known in MOOCs [17,34,35]. The studies advocated that perceived usefulness referred to whether technology could support users in completing a specific task. Thus, it reflects on how users subjectively perceive the usefulness of MOOCs to them. In this regard, perceived usefulness is a driving factor to achieve the learning objectives through a specific system Davis et al. [22] which can influence attitudes and is a direct determinant of ongoing IS use intentions. According to several studies, perceived usefulness is significant in influencing users' behaviour toward MOOCs [17,34,35]. Hence, we hypothesises that:

**Hypothesis 3 (H3).** *Perceived usefulness has a significant positive relationship with attitude toward using MOOCs.*

On the other hand, the extant literature [22] postulated that perceived usefulness is influenced by perceived ease of use where it can also influence users' attitude in using the technology. Furthermore, perceived usefulness mediates the perceived ease of use for behavioural purposes [36]. Therefore, it could be hypothesises that:

**Hypothesis 3a (H3a).** *Usefulness has a significant positive mediation on the relationship between ease of use and students' attitude.*

Wu and Chen [17] revealed that perceived usefulness is a direct determinant of continued usage intention. In this study, MOOCs' perceived usefulness refers to the belief of an individual on whether MOOCs could lead to the achievement of learning goals. Therefore, this study hypothesises that:

**Hypothesis 4 (H4).** *Perceived usefulness has a significant positive relationship with the continued use of MOOCs.*

The link between attitude and intention underlined in TAM denotes that the user's attitude serves as an assessment predisposition for behaviour. The behaviour toward the utilisation of MOOCs was perceived as the degree to which a person feels that the MOOCs are positive or negative. Moreover, previous studies have revealed that attitude is the most powerful technology use predictor [37]. In a study conducted in China, MOOCs' attitudes and perceived behavioural control (PBC) were significant factors for its use in the country [7]. Meanwhile, Alraimi et al. [36] suggested a theoretical model grounded on the information system confirmatory expectation model to examine the factors that influence the continued intention to use MOOCs'. Therefore, this study hypothesises that:

**Hypothesis 5 (H5).** *Students' attitude has a significant positive relationship with the continuance intention to use MOOCs.*

According to Joo et al. [31], perceived usefulness was positively related to MOOCs' continuity intention and on the intention to maintain MOOCs. In particular, attitude served as a crucial mediator between perceived usefulness on the intention to use MOOCs. Hence, perceived usefulness indirectly influenced the intention of using MOOCs based on the user's attitude [17]. Thus, this study hypothesises that:

**Hypothesis 5a (H5a).** *Students' attitudes act as a significant positive mediator in the relationship between usefulness and continuous intention to use MOOCs.*

*2.5. Individual–Technology Fit*

Individual–technology fit (ITF) denotes the extent of technologies' characteristics that fit individuals' desire to solve problems. The effective use of MOOCs by students depends on individual technology factors, including whether the education methods coincide with

the learning styles and correspond to MOOCs' contents. Individual interactions with the information system are usually interconnected with their adaptation to technology [38]. The technology that works to match the needs and skills of individual tasks with more experience implies that individual technology fits more easily. In addition to the powerful impact of expertise on the perceived ease of operation, experience is linked to perceived utility because experienced users effectively understand the tool's utility [9,17]. Therefore, the following hypotheses were derived:

**Hypothesis 6 (H6).** *MOOCs' individual–technology fit has a significant positive relationship with their perceived usefulness.*

**Hypothesis 7 (H7).** *MOOCs' individual–technology fit has a significant positive relationship with their perceived ease of use.*

**Hypothesis 7a (H7a).** *Perceived ease of use mediates the relationship between MOOCs' individual technology and perceived usefulness.*

*2.6. TTF*

TTF is an empirical measure that explains the extent to which IS capabilities match the users' tasks that need to be performed [39]. Several empirical studies [40,41] have demonstrated that perceived ease of use and perceived usefulness of a particular technology may serve as a foundation to ascertain whether a particular technology matches the present values. Therefore, TTF is used to evaluate user performance as it determines the factors influencing the use of technology utility and task requirements [42]. Technology is accepted by users if it matches the task. Based on the empirical results, task–technology fitness can affect the perceived ease of use and perceived usefulness whereby users perceive the tool as user-friendly and useful when it is higher [17]. Therefore, this study hypothesises that:

**Hypothesis 8 (H8).** *Task technology positively influences students' perceived usefulness of MOOCs.*

**Hypothesis 9 (H9).** *Students' perceived ease of use of MOOCs is positively influenced by TTF.*

**Hypothesis 9a (H9a).** *Ease of use mediates the relationship between TTF and perceived usefulness.*

*2.7. Openness (Cast Advantage)*

An open education community such as MOOC has concentrated over time on greater openness, enabling the community to provide visible and accessible educational materials through free access, greater choice, and flexibility [40,41]. Many students are interested in MOOCs due to their widespread and open nature compared to receiving certificates of any kind or academic credit [43]. Mohapatra and Mohanty [44] further revealed that behavioural intention to use MOOCs is significantly and positively affected by affordability. In the same vein, MOOCs' openness is the very reason to join MOOCs [45]. For example, students in Middle Eastern countries such as Saudi Arabia, the UAE, and Egypt pay for private tutoring to be able to grasp the subjects taught at higher institutions [46]. Thus, many universities offer free, open MOOCs to allow students to freely and flexibly access educational resources [17]. Nevertheless, courses available online are limited and may not meet the students' needs (quality and type). Meanwhile, some universities offer various MOOC courses at a high cost, which affects students' continuous intention to use MOOCs. Therefore, this study argues that cost advantage refers to students' perception and satisfaction with MOOCs' cost, which is commensurate with the courses' quality and variety. Based on the discussion above, this study suggests the following hypotheses:

**Hypothesis 10 (H10).** *Openness can significantly and positively affect the usefulness of using MOOCs.*

**Hypothesis 11 (H11).** *Openness can significantly and positively affect the ease of use of MOOCs.*

**Hypothesis 11a (H11a).** *The relationship between openness (course advantage) and perceived usefulness is mediated by students' perceived ease of use regarding MOOCs.*

### 2.8. Reputation

MOOCs' reputation influences users' beliefs on MOOCs [47]. In this light, an effective MOOC can draw more participation [48] because students' and lecturers' participation is dependent on their beliefs and their views on MOOCs [9]. The reputation of educational institutions is alluring to students because it serves as the perceived excellence of an institution which influences the choices of future students to attend the school [43,49]. In short, the initial assessment of the course's reputation or university can, therefore, influence the student's decision to withdraw [48]. At the early stages of a course, reputation could be a critical determiner to the student's attitude toward the programme because they lack the experience to evaluate the course based on merits [48]. This study proposes the following hypothesis:

**Hypothesis 12 (H12).** *Students' perceived usefulness toward MOOCs is positively influenced by MOOCs' reputation.*

### 2.9. Social Recognition

Recognition could significantly affect individuals' abilities and capabilities, and facilitate social interaction. Recognition allows the development of social interaction based on the in-depth understanding and awareness of self-respect, self-confidence, self-esteem, and social relationships. Thus, social interaction is influenced by recognition. Many studies have examined the different forms and patterns of social recognition, while limited studies explained the implementation of social recognition in MOOCs and their comprehensive impact on the perceived usefulness. Furthermore, the use of MOOCs among students may also be driven by an expected increase in grades, rewards, and support [38,50]. Social recognition also holds an effective role in enhancing ones' aptitudes and skills to enable social interaction. A deep understanding of self-respect, self-confidence, self-esteem, and evolving relations in a society is created by social recognition [17]. Social interaction can be generated if recognition is established, which is considered to be a key factor in learning speeches. Although the available literature highlighted the different social recognition patterns, it remains insufficient to help recognise the effects of social recognition in accepting MOOCs. However, Khan et al. [9] added that MOOCs' use can be increased by increasing expected levels, reward structure, and institutional support. This study, therefore, proposes the following hypothesis:

**Hypothesis 13 (H13).** *The perceived usefulness of MOOCs is influenced by social recognition.*

### 2.10. Social Influence

Studies on information systems revealed that individuals tend to adopt specific technologies based on the view of others rather than based on their preferences [51]. As such, Hsu and Lu [52] highlighted the impact of social influence on IT users' acceptability and the empirical support as user behaviour drivers. Several studies have also revealed that people are motivated to follow the beliefs of others to strengthen their relationships with the members of the group [53,54].

This study considers social influence based on the extent to which other users explicitly agree and encourage participation in MOOCs [17]. A person is more willing to use MOOCs or increase the current use and future use of MOOCs if they notice others using

and benefiting from them. Similarly, this study expects the moderation effects of social influence on the student's perception of MOOCs' use. Therefore, this study proposes the following hypotheses:

**Hypothesis 14 (H14).** *The perceived usefulness of MOOCs is positively influenced by social influence.*

**Hypothesis 15 (H15).** *The attitude toward using MOOCs is positively influenced by social influence.*

**Hypothesis 15a (H15a).** *Students' perceived usefulness of MOOCs mediates the relationship between social influence and students' attitude toward the use of MOOCs.*

### 2.11. Arab Language Support

Language study is a vital and important skill in today's competitive international environment. However, most MOOCs are conducted in English (6287), whereby only 126 MOOCs use Arabic as the medium of instruction [55]. This may affect user's attitude toward the continuous use of MOOCs, particularly in learning environments that are predominantly localised. Thus, recently, several educational institutions such as universities provide both English and Arabic MOOC platforms, and both languages exclusively provide the courses. Joseph and Nath [56] advocated the use of the participants' native language in delivering MOOCs based on the students' cultural background/context. Hence, MOOC participants who are enrolled in foreign language courses might face difficulty understanding the course content due to their low language proficiency [57]. According to Liu [58], the most difficult obstacle for students without English skills is the English courses, and they are less interested in taking the lessons. Since Muslims constitute the globe's second-largest population, and many Muslims speak Arabic, Arabic is the official language of Islam. That said, the effects of MOOC languages in the acceptance and continuation of MOOCs have not been investigated [59]. Therefore, we hypothesise that:

**Hypothesis 16 (H16).** *Perceived ease of use of MOOCs is influenced by the use of the Arabic language.*

**Hypothesis 17 (H17).** *Perceived usefulness of MOOCs is influenced by the use of the Arabic language.*

**Hypothesis 17a (H17a).** *Students' perceived ease of use mediates the relationship between MOOCs' Arabic language and students' perceived usefulness.*

### 2.12. MOOCs' Accreditation

MOOCs' accreditation is a process of providing credit or recognition for students after completing an MOOC. MOOC participants who enrolled and completed an MOOC usually do not receive any certification other than a letter of completion [11]. This practice differs from conventional courses where students receive a certificate upon completing a degree. Although some MOOC providers do provide certification [31], a more standard accreditation is required. Ghislandi [60] argued that accreditation is a complex but compulsory process in many countries, where academic institutions are subjected to accreditation by government or private accreditation agencies [61]. In most countries, the official accreditation agency is the respective Ministry of Education. Meanwhile, Aldahdouh and Osório [62] claimed that some MOOC providers failed the accreditation process.

MOOCs' accreditation is quite different, as students are usually enrolled in a single course. While some providers offer related courses through MOOCs [62], MOOC students are not required to follow a study plan. As a result, only the course needs to be an accred-

ited degree. According to on-campus students, accreditation is one of their primary concerns regarding MOOCs [63]. However, Aldahdouh and Osório [62] revealed that 57% of the EdX course participants do have the intention to be certified, even though formal accreditation in MOOCs is less emphasised. Therefore, educators facilitate MOOCs voluntarily where the MOOCs' participants can enrol themselves in search of knowledge rather than of certification [64]. However, Almuhanna [65] discovered that students enrol in MOOCs' courses that are accredited and certified because they are recognised and useful during job hunting. Therefore, this study hypothesises:

**Hypothesis 18 (H18).** *MOOCs' accreditation and certification positively affect students' perceived usefulness.*

## 3. Methodology

### 3.1. Data Collection

This study aims to determine the motivational and technological factors that influence the continuance intention to use MOOCs among the students of UAE. A descriptive research design driven by quantitative methodology was adopted. The respondents were contacted, the purpose of the study was explained to them, and they were assured of confidentiality. The researchers then assigned representatives and directly distributed questionnaire forms to participants. The students were requested to distribute the survey forms randomly to minimise bias. Extra procedures were adopted to mitigate self-reporting bias; for example, the questionnaire was online and randomly distributed, variables were not titled, the respondents were assured of anonymity and no personal information was requested, and participation was completely voluntary. In addition, no data were collected from anyone under 16 years old. Therefore, the ethics committee at the Universiti Teknikal Malaysia Melaka ruled that no formal ethics approval was required, and the participants provided informed consent. A total of 1690 students are using MOOCs at the university. Based on Krejcie and Morgan Table, the sample size was determined to be roughly 320 respondents. Therefore, questionnaires were distributed through an online survey to 491 students in August 2020. The use of online questionnaires reduced the possibility of missing data because respondents were required to answer all items in the questionnaires before submitting their responses. Three hundred forty-nine questionnaires were returned after three weeks. This indicates that the initial response rate is 67 percent. Of the 491 responses, 315 were deemed valid for analysis after excluding 15 questionnaires because they contained unengaged responses, as they recorded a standard deviation value of zero. Each item in the questionnaire was first translated into Arabic and retranslated into English to ensure consistency.

### 3.2. Measurement and Scales

All of the measurement items in this study were derived from previous valid instruments. Five items for continuance intention to use MOOCs (CIUU) were adopted from [42]; four items for Individual–Technology Fit (ITF) from [17]; and three items for TTF from [66]. Meanwhile, five items for openness (OP) were adopted from [44,46]; five items for programme reputation (PR) from [44]; three items for social recognition (SR) from [38]; four items for social influence (SI) from [67]; three items for Arabic language use (ALS) from [59]; five items for course accreditation and certification (CCA) from [65]; seven items for pperceived ease of use (PEU) from [68]; six items for perceived usefulness (UUL) from [68]; and four items for attitude (ATU) from [34].

### 3.3. Data Analysis Method

The partial least squares structural equation modelling (PLS-SEM) was employed to measure the complex cause–effect relationship models with latent variables [69]. The au-

thors also added that comparing covariance-based structural equation modelling methodologies are appropriate to evaluate higher-order constructs and sophisticated conceptual models with mediation effects [69]. The PLS-SEM technique using Smart-PLS was suitable to examine the cause–effect linkages presented in this study model because the study sample surpassed 100 (n = 315).

*3.4. Data Analysis*

3.4.1. Demographic Characteristics

The demographic profiles of the respondents in this study are summarised in Table 1. More males participated in this study (81.5%) than female respondents (18.5%). Most of the respondents (45%) were from the School of Business and Quality Management, while 32% were from the School of E-education, and 23% were from the School of Health and Environment Studies. Most of the respondents have between 1–2 years of experience (45%), 28% have 2–3 years of experience, and 27% have less than 1 year of experience. A majority of the respondents used both xMOOCs and cMOOCs (57%), followed by xMOOCs only (28%) and cMOOCs (15%).

**Table 1.** Demographic characteristics.

| Gender | N | % |
|---|---|---|
| Female | 58 | 18.5 |
| Male | 257 | 81.5 |
| Total | 315 | 100 |
| School | | |
| School of Business and Quality Management | 141 | 45 |
| School of Health and Environment Studies | 74 | 23 |
| School of E-education | 100 | 32 |
| Total | 315 | 100 |
| Experience | | |
| Less than 1 year | 85 | 27 |
| 1–2 years | 141 | 45 |
| 2–3 years | 89 | 28 |
| Total | 315 | 100 |
| Type of MOOCs | | |
| xMOOCs | 87 | 28 |
| cMOOCs | 49 | 15 |
| Both | 179 | 57 |
| Total | 315 | 100 |

3.4.2. Validity and Reliability

In the first stage of SEM, construct reliability, indicator reliability, convergent validity, and discriminant validity of the identified constructs were assessed. Composite reliability (CR) and Cronbach's alpha were used to determine the construct dependability (CA). The construct is rendered reliable if the CR value is greater than 0.07. The CR values obtained for this study were greater than 0.07, indicating sufficient CA (Table 2). The indicator dependability was then tested using CA (CA > 0.07). Based on the results, the CA for all variables was acceptable. Meanwhile, average variance extracted (AVE) was used to determine the constructs' convergent validity where the value should be greater than 0.50 [70]. The convergent validity of the constructs for this study was evident because all constructs yielded significant AVE. Table 2 lists the values of CA, CR, and AVE.

**Table 2.** Reliability and validity.

| Construct | Mean | SD | CA | CR | AVE |
|---|---|---|---|---|---|
| ALS | 4.603 | 1.825 | 0.676 | 0.822 | 0.607 |
| ATU | 4.935 | 1.736 | 0.778 | 0.858 | 0.604 |
| CCA | 4.689 | 1.79 | 0.753 | 0.835 | 0.504 |
| CIUU | 4.904 | 1.749 | 0.806 | 0.865 | 0.562 |
| ITF | 4.713 | 1.7704 | 0.895 | 0.922 | 0.704 |
| OP | 4.825 | 1.758 | 0.785 | 0.874 | 0.699 |
| PEU | 4.874 | 1.699 | 0.887 | 0.917 | 0.689 |
| PR | 4.653 | 1.784 | 0.888 | 0.918 | 0.691 |
| SI | 4.535 | 1.7915 | 0.894 | 0.926 | 0.758 |
| SR | 4.943 | 1.722 | 0.746 | 0.854 | 0.661 |
| TTF | 4.748 | 1.71 | 0.845 | 0.906 | 0.762 |
| UUL | 5.039 | 1.6915 | 0.908 | 0.929 | 0.686 |

The discriminant validity of the variables was assessed using three methods, namely, the Fornell and Lacker criterion, cross-loading, and the Heterotrait–Monotrait ratio (HTMT) [69]. The discriminant validity was assessed based on the comparison of the square root of AVE retrieved from each concept with the correlation between constructs using the Fornell–Lacker criterion. The cross-loading approach then assesses the construct's outer loading, which should be greater than the associated construct loading to indicate appropriate discriminant validity.

Table 3 lists the findings of the Fornell–Lacker and HTMT tests, while Table 4 presents the results of cross-loading. According to Kline et al. [71], values above 0.85 imply that the measurement has good discriminant validity. Therefore, the discriminant validity of the constructs was determined as all loadings of the constructs were higher than the other constructs. Table 4 presents the results of cross-loading, which indicates that there is a high correlation between items of the same construct and a very weak correlation between items of a different construct. Finally, the Fornell–Lacker criterion was used to verify discriminant validity, which demonstrated substantial relationships between the constructs.

**Table 3.** Discriminant validity.

| | ALS | ATU | CCA | CIUU | ITF | OP | PEU | PR | SI | SR | TTF | UUL |
|---|---|---|---|---|---|---|---|---|---|---|---|---|
| **ALS** | 0.779 | | | | | | | | | | | |
| **ATU** | 0.347 | 0.777 | | | | | | | | | | |
| **CCA** | 0.387 | 0.577 | 0.71 | | | | | | | | | |
| **CIUU** | 0.401 | 0.673 | 0.628 | 0.75 | | | | | | | | |
| **ITF** | 0.251 | 0.417 | 0.256 | 0.403 | 0.839 | | | | | | | |
| **OP** | 0.391 | 0.472 | 0.413 | 0.425 | 0.406 | 0.836 | | | | | | |
| **PEU** | 0.365 | 0.534 | 0.372 | 0.536 | 0.487 | 0.47 | 0.83 | | | | | |
| **PR** | 0.233 | 0.394 | 0.237 | 0.325 | 0.435 | 0.338 | 0.298 | 0.831 | | | | |
| **SI** | 0.353 | 0.584 | 0.385 | 0.456 | 0.492 | 0.388 | 0.414 | 0.502 | 0.871 | | | |
| **SR** | 0.43 | 0.331 | 0.304 | 0.354 | 0.458 | 0.494 | 0.461 | 0.202 | 0.274 | 0.813 | | |
| **TTF** | 0.336 | 0.489 | 0.34 | 0.416 | 0.712 | 0.412 | 0.482 | 0.461 | 0.518 | 0.48 | 0.873 | |
| **UUL** | 0.343 | 0.644 | 0.444 | 0.583 | 0.621 | 0.555 | 0.711 | 0.473 | 0.604 | 0.516 | 0.636 | 0.828 |

**Table 4.** Loadings and cross-loading.

| | ALS | ATU | CCA | CIUU | ITF | OP | PEU | PR | SI | SR | TTF | UUL |
|---|---|---|---|---|---|---|---|---|---|---|---|---|
| MOOCs are provided in the Arabic language | 0.836 | 0.246 | 0.299 | 0.269 | 0.227 | 0.338 | 0.349 | 0.228 | 0.252 | 0.302 | 0.289 | 0.27 |

| | | | | | | | | | | | | |
|---|---|---|---|---|---|---|---|---|---|---|---|---|
| Many internet sources of information are in Arabic. | 0.791 | 0.336 | 0.408 | 0.321 | 0.188 | 0.367 | 0.262 | 0.178 | 0.356 | 0.441 | 0.278 | 0.28 |
| Access to the verity of information in the Arabic language. | 0.705 | 0.231 | 0.191 | 0.364 | 0.167 | 0.195 | 0.231 | 0.128 | 0.218 | 0.265 | 0.213 | 0.254 |
| MOOCs, in my opinion, are a good idea. | 0.292 | 0.775 | 0.412 | 0.502 | 0.279 | 0.318 | 0.384 | 0.317 | 0.511 | 0.19 | 0.366 | 0.412 |
| MOOCs, in my opinion, are beneficial. | 0.342 | 0.824 | 0.517 | 0.524 | 0.346 | 0.407 | 0.441 | 0.221 | 0.436 | 0.322 | 0.435 | 0.523 |
| In MOOCs, I actively participate in a variety of sorts of debates and reviews. | 0.219 | 0.675 | 0.473 | 0.535 | 0.259 | 0.314 | 0.396 | 0.426 | 0.402 | 0.255 | 0.289 | 0.506 |
| I am satisfied with using MOOCs. | 0.224 | 0.826 | 0.387 | 0.526 | 0.405 | 0.419 | 0.433 | 0.263 | 0.465 | 0.256 | 0.424 | 0.553 |
| MOOCs offer us a flexible schedule to finish our course. | 0.198 | 0.394 | 0.685 | 0.449 | 0.163 | 0.33 | 0.263 | 0.163 | 0.254 | 0.125 | 0.201 | 0.299 |
| I can easily transform MOOCs' credit. | 0.406 | 0.447 | 0.753 | 0.455 | 0.187 | 0.288 | 0.285 | 0.112 | 0.288 | 0.2 | 0.276 | 0.336 |
| MOOCs' programmes are certified and accredited in my university. | 0.283 | 0.426 | 0.758 | 0.441 | 0.207 | 0.28 | 0.303 | 0.059 | 0.222 | 0.233 | 0.292 | 0.32 |
| MOOCs are beneficial for professional development, which is essential for job placement. | 0.228 | 0.455 | 0.747 | 0.487 | 0.167 | 0.291 | 0.278 | 0.283 | 0.331 | 0.236 | 0.239 | 0.377 |
| MOOCs provided us a statement of accomplishment when we finished our study. | 0.268 | 0.298 | 0.595 | 0.394 | 0.199 | 0.294 | 0.17 | 0.239 | 0.275 | 0.321 | 0.188 | 0.214 |
| In the future, I plan to continue using MOOCs. | 0.325 | 0.547 | 0.425 | 0.741 | 0.316 | 0.323 | 0.42 | 0.224 | 0.255 | 0.272 | 0.358 | 0.424 |
| I will insist on using MOOCs to study the courses I registered. | 0.361 | 0.401 | 0.455 | 0.75 | 0.279 | 0.315 | 0.327 | 0.205 | 0.312 | 0.317 | 0.326 | 0.395 |
| Rather than adopting any other method, I intend to continue using the MOOCs system. | 0.334 | 0.487 | 0.493 | 0.719 | 0.298 | 0.293 | 0.408 | 0.198 | 0.28 | 0.282 | 0.336 | 0.417 |
| In the future, I plan to use the MOOCs system regularly. | 0.268 | 0.58 | 0.544 | 0.812 | 0.371 | 0.402 | 0.484 | 0.242 | 0.44 | 0.309 | 0.343 | 0.53 |
| I would recommend the use of MOOCs to others. | 0.23 | 0.479 | 0.428 | 0.724 | 0.228 | 0.243 | 0.345 | 0.354 | 0.41 | 0.143 | 0.189 | 0.401 |
| I am capable of completing MOOCs courses on my own and consciously. | 0.216 | 0.334 | 0.173 | 0.33 | 0.856 | 0.354 | 0.401 | 0.357 | 0.39 | 0.345 | 0.551 | 0.488 |
| MOOCs' features are sufficient to complete my study. | 0.207 | 0.282 | 0.248 | 0.305 | 0.84 | 0.354 | 0.363 | 0.346 | 0.34 | 0.388 | 0.549 | 0.495 |
| MOOCs' system fits well with the way to enhance my learning. | 0.215 | 0.42 | 0.247 | 0.355 | 0.821 | 0.321 | 0.428 | 0.352 | 0.464 | 0.352 | 0.652 | 0.573 |
| I strive to obtain accolades for great MOOC performance. | 0.214 | 0.379 | 0.219 | 0.368 | 0.853 | 0.364 | 0.473 | 0.386 | 0.41 | 0.442 | 0.598 | 0.544 |
| MOOCs' system fits well with the way I like to strengthen my learning skills. | 0.201 | 0.319 | 0.181 | 0.325 | 0.825 | 0.307 | 0.364 | 0.384 | 0.453 | 0.391 | 0.629 | 0.495 |
| I am free to enrol in any course with no prerequisites. | 0.264 | 0.327 | 0.308 | 0.323 | 0.307 | 0.826 | 0.395 | 0.331 | 0.297 | 0.38 | 0.337 | 0.442 |
| I have permission to access and use the course materials and resources. | 0.32 | 0.436 | 0.418 | 0.382 | 0.334 | 0.826 | 0.371 | 0.202 | 0.348 | 0.437 | 0.38 | 0.449 |

| | | | | | | | | | | | | |
|---|---|---|---|---|---|---|---|---|---|---|---|---|
| I will be able to apply the course materials to my work. | 0.39 | 0.418 | 0.314 | 0.36 | 0.374 | 0.856 | 0.411 | 0.312 | 0.329 | 0.423 | 0.32 | 0.499 |
| It is easy to learn how to use MOOCs. | 0.289 | 0.406 | 0.286 | 0.412 | 0.376 | 0.369 | 0.814 | 0.284 | 0.313 | 0.346 | 0.37 | 0.596 |
| It is easy to master the use of MOOCs. | 0.36 | 0.446 | 0.304 | 0.459 | 0.407 | 0.422 | 0.834 | 0.215 | 0.393 | 0.43 | 0.381 | 0.605 |
| I felt free to ask questions throughout this course. | 0.3 | 0.469 | 0.348 | 0.461 | 0.416 | 0.378 | 0.85 | 0.267 | 0.336 | 0.401 | 0.425 | 0.599 |
| The instructor responded to my questions in a timely manner. | 0.262 | 0.469 | 0.292 | 0.44 | 0.473 | 0.363 | 0.816 | 0.269 | 0.36 | 0.367 | 0.471 | 0.554 |
| Using MOOCs leads to easy access to resources from anywhere and at any time. | 0.302 | 0.423 | 0.311 | 0.45 | 0.346 | 0.415 | 0.834 | 0.203 | 0.313 | 0.367 | 0.349 | 0.597 |
| The MOOC platform has a good reputation and offers courses that I am interested in. | 0.191 | 0.336 | 0.22 | 0.225 | 0.35 | 0.327 | 0.224 | 0.828 | 0.413 | 0.186 | 0.382 | 0.38 |
| MOOCs' partners in our university have a good reputation. | 0.177 | 0.332 | 0.203 | 0.307 | 0.351 | 0.218 | 0.226 | 0.839 | 0.374 | 0.114 | 0.356 | 0.382 |
| It is easy to interact with professors and lecturers through MOOCs. | 0.184 | 0.276 | 0.147 | 0.241 | 0.33 | 0.25 | 0.251 | 0.833 | 0.382 | 0.128 | 0.346 | 0.367 |
| MOOCs' courses have high standards in voice and image. | 0.196 | 0.362 | 0.201 | 0.272 | 0.418 | 0.339 | 0.281 | 0.809 | 0.48 | 0.206 | 0.438 | 0.436 |
| MOOCs platform offers courses of excellent quality. | 0.217 | 0.322 | 0.211 | 0.304 | 0.35 | 0.263 | 0.251 | 0.845 | 0.427 | 0.196 | 0.384 | 0.392 |
| The opinions of other participants concerning MOOCs motivate me to use them. | 0.271 | 0.455 | 0.29 | 0.352 | 0.401 | 0.322 | 0.323 | 0.415 | 0.838 | 0.175 | 0.374 | 0.501 |
| The opinions of other participants about MOOCs have an impact on my capacity to use them. | 0.28 | 0.511 | 0.348 | 0.392 | 0.442 | 0.349 | 0.385 | 0.429 | 0.858 | 0.273 | 0.437 | 0.534 |
| MOOC credentials must be verified by colleges. | 0.359 | 0.47 | 0.316 | 0.389 | 0.443 | 0.301 | 0.372 | 0.47 | 0.887 | 0.267 | 0.486 | 0.511 |
| The belief of other participants toward MOOCs influences my decision to use them. | 0.318 | 0.587 | 0.381 | 0.448 | 0.428 | 0.374 | 0.362 | 0.437 | 0.899 | 0.236 | 0.499 | 0.555 |
| It is imperative for employers to adopt MOOCs as on-the-job training. | 0.301 | 0.226 | 0.204 | 0.256 | 0.384 | 0.406 | 0.37 | 0.156 | 0.215 | 0.836 | 0.363 | 0.418 |
| It is important for MOOCs' quality to be appreciated and accepted by others. | 0.393 | 0.262 | 0.289 | 0.224 | 0.392 | 0.388 | 0.365 | 0.193 | 0.21 | 0.775 | 0.383 | 0.346 |
| It i essential that MOOC credentials are verified by colleges. | 0.364 | 0.314 | 0.256 | 0.363 | 0.35 | 0.41 | 0.389 | 0.151 | 0.24 | 0.827 | 0.423 | 0.476 |
| MOOCs are appropriate for my educational needs. | 0.267 | 0.369 | 0.234 | 0.296 | 0.597 | 0.316 | 0.368 | 0.393 | 0.35 | 0.391 | 0.848 | 0.46 |
| Using MOOCs fits with my educational practice. | 0.341 | 0.471 | 0.37 | 0.397 | 0.656 | 0.434 | 0.441 | 0.393 | 0.47 | 0.446 | 0.906 | 0.563 |
| It i easy to figure out which tool to utilise in MOOCs. | 0.269 | 0.431 | 0.276 | 0.384 | 0.61 | 0.325 | 0.444 | 0.421 | 0.515 | 0.418 | 0.864 | 0.622 |
| MOOCs are ideal for assisting me in completing online courses. | 0.328 | 0.552 | 0.394 | 0.485 | 0.477 | 0.447 | 0.63 | 0.345 | 0.493 | 0.437 | 0.555 | 0.832 |

| MOOCs make it easy to convert learning content into specific learning objects (knowledge and/or skill). | 0.262 | 0.549 | 0.352 | 0.495 | 0.529 | 0.442 | 0.583 | 0.388 | 0.503 | 0.397 | 0.542 | 0.873 |
| Learning in MOOCs is easier than learning in face-to-face classes. | 0.304 | 0.57 | 0.42 | 0.513 | 0.545 | 0.51 | 0.637 | 0.413 | 0.551 | 0.437 | 0.52 | 0.838 |
| Using MOOCs would enable me to accomplish tasks more effectively. | 0.214 | 0.521 | 0.35 | 0.504 | 0.525 | 0.447 | 0.578 | 0.46 | 0.519 | 0.389 | 0.496 | 0.812 |
| Using MOOCs would enhance my effectiveness in learning. | 0.31 | 0.471 | 0.343 | 0.448 | 0.492 | 0.462 | 0.543 | 0.35 | 0.497 | 0.467 | 0.531 | 0.809 |
| I would recommend this course to friends/colleagues. | 0.288 | 0.533 | 0.343 | 0.448 | 0.519 | 0.451 | 0.557 | 0.392 | 0.435 | 0.44 | 0.518 | 0.805 |

*3.5. Path Analysis*

Based on Table 5, the structural model results revealed that the causal relationship between individual task fit and perceived ease of use was statically significant. However, the causal relationship between individual task fit and perceived usefulness was insignificant. Meanwhile, the relationships between task technology fit and both perceived ease of use and perceived usefulness were significant. On the other hand, openness yielded a significant positive relationship with ease of use but revealed no relationship with perceived usefulness. The relationship between reputation and perceived usefulness was statically significant, and social influence and attitude toward MOOCs were also positively significant. Similarly, the relationship between social recognition and usefulness was significantly positive. Likewise, the relationships between social influence and usefulness, and social recognition and attitude were also significantly positive. The use of the Arabic language positively and significantly influenced ease of use. While there was also a significant positive relationship between course certificate accreditation and perceived usefulness. Contrarily, the relationship between the use of the Arabic language and perceived usefulness was insignificant. However, the relationship between perceived ease of use and perceived usefulness was significantly positive, like perceived ease of use and attitude, which was also significant. Perceived usefulness also positively and significantly influenced student attitude and continuance intention to use, and there was a significant relationship between attitude and continuance intention to use. Next, the effect size ($f^2$) for this study is tabulated in Table 5, ranging from 0.000 to 0.072, indicating that all constructs generated a small effect size on the students' continuance intention to use MOOCs.

As for the blindfolding process, Hair et al. [72] demonstrated how construct values are well-observed by reconstructing parameter estimations. In this study, only the endogenous constructs with reflected indicators were blindfolded. The model's predictive relevance ($Q^2$) was calculated collectively by considering all components at an individual level (single factor) and is presented in Table 5. The blindfolding method yielded a predictive relevance of 0.714, indicating the integration of variables for students' continued intention to use MOOCs.

**Table 5.** Path coefficients.

| Path | Path Coefficient | S.E | t-Value | *p*-Value |
|---|---|---|---|---|
| PEU→UUL | 0.381 | 0.053 | 7.195 | 0.000 |
| PEU→ATU | 0.163 | 0.067 | 2.422 | 0.015 |
| UUL→ATU | 0.340 | 0.075 | 4.553 | 0.000 |
| UUL→CIUU | 0.256 | 0.075 | 3.422 | 0.001 |
| ATU→CIUU | 0.508 | 0.067 | 7.622 | 0.000 |
| ITF→UUL | 0.091 | 0.094 | 1.613 | 0.067 |
| ITF→PEU | 0.232 | 0.065 | 3.589 | 0.000 |
| TTF→UUL | 0.128 | 0.059 | 2.171 | 0.030 |

| | | | | |
|---|---|---|---|---|
| TTF→PEU | 0.162 | 0.069 | 2.350 | 0.019 |
| OP→UUL | −0.084 | 0.087 | 1.421 | 0.072 |
| OP→PEU | 0.248 | 0.061 | 4.088 | 0.000 |
| PR→UUL | 0.089 | 0.039 | 2.276 | 0.023 |
| SR→UUL | 0.108 | 0.045 | 2.399 | 0.016 |
| SI→UUL | 0.205 | 0.050 | 4.067 | 0.000 |
| SI→ATU | 0.311 | 0.052 | 5.947 | 0.000 |
| ALS→PEU | 0.155 | 0.053 | 2.933 | 0.003 |
| ALS→UUL | −0.083 | 0.037 | 1.753 | 0.066 |
| CCA→UUL | 0.083 | 0.039 | 2.160 | 0.031 |

### 3.6. Mediating Effects

As shown in Table 6, perceived usefulness exhibited a partial mediating effect on the relationship between perceived ease of use and attitude toward the use of MOOCs. The coefficient of perceived usefulness toward the attitude to using MOOCs was recorded at 0.386 (*p*-value = 0.0001). This study also demonstrated that the students' attitudes partially mediated the relationship between perceived usefulness and students' continuance intention to use MOOCs. The b coefficient of perceived usefulness in continuance intention to use MOOCs was estimated at 0.330 (*p*-value = 0.0001).

On the other hand, perceived ease of use fully mediated the relationship between individual–technology fit and perceived usefulness. The direct effect of individual–technology fit and perceived usefulness yielded a zero value between the lower and upper confidence intervals. Moreover, VAF does not work since the direct effects have different signs, and when a direct effect is negative, the VAF value is 100% [72]. Meanwhile, the indirect effect of perceived ease of use in the relationship between individual–technology fit and perceived usefulness was 0.252, and the direct effect (individual–technology fit and perceived usefulness) was 0.089. Hence, the VAF value was estimated to be 100%. It was concluded that the perceived ease of use fully mediated individual–technology fit and perceived usefulness because the VAF value was greater than 80%.

In addition to that, perceived ease of use partially mediated the relationship between technology–tasks fit and perceived usefulness. The b coefficient of technology–tasks fit in perceived usefulness was recorded at 0.254 (*p*-value = 0.001).The indirect effects of perceived ease of use in the relationship between MOOCs' openness (course advantage) and perceived usefulness were significant at a coefficient of 0.231 (*p*-value = 0.002), and the direct effect of MOOCs' openness (course advantage) on perceived usefulness was 0.091. In this study, perceived usefulness was identified to have partially mediated the relationship between social influence and students' attitudes toward MOOCs. The coefficient of social influence on the attitude toward MOOCs was recorded at 0.278 (*p*-value = 0.0001). Finally, perceived ease of use fully mediated the relationship between the use of the Arabic language and perceived usefulness. The indirect effect of perceived ease of use in the relationship between the use of the Arabic language in MOOCs (courses and platform) and perceived usefulness was 0.221 (*p*-value = 0.0004), whereas the direct effect between the use of Arabic language and perceived usefulness was 0.076.

**Table 6.** Path coefficients for indirect effect.

| Path | Path Coefficient | 2.50% | 97.50% | t-Statistics | p-Values | Decision |
|---|---|---|---|---|---|---|
| PEU→UUL→ATU | 0.386 | 0.288 | 0.49 | 7.45 | 0.000 | Accepted |
| UUL→ATU→CIUU | 0.330 | 0.236 | 0.432 | 6.545 | 0.000 | Accepted |
| ITF→PEU→UUL | 0.252 | 0.162 | 0.340 | 5.548 | 0.000 | Accepted |
| TTF→PEU→UUL | 0.254 | 0.182 | 0.338 | 6.374 | 0.001 | Accepted |
| OP→PEU→UUL | 0.231 | 0.242 | 0.421 | 4.98 | 0.002 | Accepted |

| | | | | | | |
|---|---|---|---|---|---|---|
| SI→UUL→ATU | 0.278 | 0.203 | 0.36 | 6.931 | 0.000 | Accepted |
| ALS→PEU→UUL | 0.221 | 0.245 | 0.446 | 3.91 | 0.004 | Accepted |

## 4. Discussion

In line with technological development enhanced by educational competition, massive open online courses (MOOCs) play a major role in the development of education at the level of institutions and students. At the institutional level, MOOCs help to support the competitive advantage of institutions by providing educational services at the lowest costs and that are the most effective. On the other hand, at the students' level, MOOCs provide students good-quality and lower-cost courses with the flexibility of accessing the contents and interactive tools for learning. This study aimed to assess the influence of behavioural and technological factors on MOOCs' continuation intentions among the students. This study verified the significant relationship between the perceived ease of using MOOCs and their perceived usefulness [31,73]. The results indicated that high perceived usefulness is evident when the platform could be easily used by users, which is supported by study [31]. As a result, we believe students found the MOOCs' online learning system to be simple to use and beneficial to their learning. Because MOOC learners have a wide range of technical skills and ages, it is vital to build learner-friendly features so that they do not encounter technical issues during their initial use and believe MOOCs are effective for their learning activities and goals. In addition, this study revealed that MOOCs' perceived ease of use among students is not positively associated with an improvement in their attitude toward MOOCs. The friendly use of MOOCs influence students' perception and attitude to use MOOCs in their education. This study is consistent with the existing research [74]. Contrarily, the findings from this study contradicted Wu and Chen's [17] by concluding that the ease of use does not significantly affect students' attitudes to use MOOCs. However, this study revealed that perceived usefulness can positively influence students' attitudes toward MOOCs' usage. This indicates that the value of MOOCs that students can gain through MOOCs influences their attitude to use MOOCs, which influences MOOCs' stability in the long run. For example, students' perception of MOOCs' usefulness in improving learning performance through accomplishing tasks more effectively influences their attitude toward MOOCs. Therefore, the MOOCs' providers and developers have to rapidly improve MOOCs' technology and functions that facilitate students' tasks effectively and meet the rapid technological change. Our finding confirms studies by [17,59].

In addition, we also discovered that perceived usefulness can positively influence students' continuance intention to use MOOCs. The perceived usefulness in this study significantly indicated long-term intention to use MOOCs. This is because the MOOCs that are appreciated by students in terms of their effectiveness in performing the tasks compared to other educational technology attract students' intention to use them consistently. The findings of this study are in line with the findings of Wu and Chen [17], who observed that it was the antecedent of user continuance intention with regards to MOOCs. Additionally, MOOCs' practicability and usefulness can also effectively influence their usage behaviour. On the other hand, this study also revealed that attitude can positively and significantly influence students' continuance intention to use MOOCs. This shows that attitude plays an important role in the growth and development of MOOCs.

In line with previous literature [6,37], attitude was perceived as an important predictor toward the continuous intention to use MOOCs. However, a user with a poor attitude is likely to use MOOCs and then abandon them. Therefore, universities implementing MOOCs in the lodging service must focus on factors which can influence students' attitudes and their continuous use.

### 4.1. Relationship between TTF and TAM

Individual–technology fit positively and significantly influenced perceived usefulness. Moreover, the perception of users about the flexibility and efficiency of MOOCs' technology also influences their use [9,17]. In construct, this study revealed that the relationship between individual–technology fit and perceived usefulness was insignificant, in line with Wu and Chen's [17] findings. This difference could be due to individual–technology fit being more likely to influence ease of use than perceived usefulness. Moreover, this study also discovered that task–technology fitness can affect the perceived ease of use and perceived usefulness. When the fitness score is higher, users perceive the tool to be easy to operate and useful for a particular task [17]. The TTF that influences students' behaviour toward MOOCs may have technological features affecting the effectiveness of online learning. Consequently, it also influences MOOCs' perceived utility by matching the tasks and technology [9,54].

### 4.2. MOOCs' Features

This study revealed a significant and positive relationship between openness (course advantage) and perceived ease of use, in line with that of [44]. In short, course affordability influences the use of MOOCs. However, the relationship between openness (course advantage) and perceived usefulness was insignificant. Although the cost affordability of MOOCs may encourage enrolment of students, the perceived usefulness depends on the quality of the course. Even so, this study also indicated that MOOCs' reputation has a positive relationship with students' perceived usefulness. Reputed systems also produce faster response times and a large number of responses per post, as well as differences in how students ask questions. The findings of this study are consistent with a study by Al-raimi et al. [36], in which reputation influences users' perceived usefulness toward MOOCs. It is important for the UAE's educational platforms to ensure the popularity of their MOOCs and platforms in processes and materials.

### 4.3. Social Factors

Social recognition can positively influence perceived usefulness and attitude toward MOOCs [17,36]. The importance of social recognition lies in its effects on social interaction based on the awareness of self-confidence, self-respect, self-esteem, and the relationship with others within the society. Hence, a person is motivated to follow others' beliefs to strengthen the relationships with the other members of the group in explaining the conceptual reasoning underlying the link. The UAE's educational institutions have to improve their delivery quality to be recognised, as it influences the long-term stability for their MOOCs' programmes.

### 4.4. Context Factors

According to Al-Shami et al. [74], there are some barriers to MOOCs' continuous use, including language problems. This issue is critical in many countries, especially the Arab world [11]. Based on this study, the use of the Arabic language in MOOCs can positively influence ease of use [59]. However, the relationship between the Arabic language and perceived usefulness was insignificant, probably because the perceived usefulness of the Arabic language in facilitating MOOCs depends on the quality of the overall MOOCs and their interaction and information. This study further indicated that course accreditation through Course Certificate Advantage has a significantly positive relationship with perceived usefulness, in which providing certificate and accreditation for MOOCs can influence students' usefulness [62]. MOOCs that are accredited and certified are useful for recognition and to land jobs [62].

*4.5. Indirect Effect*

In this study, students' perceived usefulness mediated the relationship between perceived ease of use and attitude toward MOOCs. A platform that is easy to use will affect a learner's preferences, while a more-complex platform could receive user resistance. When users cannot navigate through the platform easily, they tend to stop using it or try other, less-complex platforms that require less effort to use [31,62]. The findings also indicated that attitude can partially mediate the relationship between usefulness and continuance intention to use MOOCs [36,75]. Moreover, the students' belief that using MOOCs could improve their performance has a positive effect on their continuous intention to use the platform. These results led scholars and the UAE's MOOCs' providers to understand the importance of highlighting MOOCs' useful aspects because it could influence students' attitudes toward using MOOCs, which subsequently impacts the postadoptive behaviour (continuance intention).

Thirdly, this study also demonstrated that perceived ease of use fully mediated the relationship between individual–technology fit and usefulness. In brief, the flexibility of a platform facilitates the usage of MOOCs, in which students can determine the volubility of the platform contents, consequently influencing their perceived usefulness. Next, perceived ease of use also mediated the relationship between tasks–technology fit and usefulness, further indicating that when technology supports the task at hand, students tend to use the same technology for such tasks [17]. Fifth, perceived ease of use was deemed as a mediator between openness (cost advantage) and perceived usefulness. Although many MOOCs are open, the dropout rate from the courses is very high, hence indicating that the cost, especially in a rich country such as the UAE, is not an issue compared to the quality of MOOCs' contents. Therefore, students appreciate the usefulness of the MOOCs after accumulated experiences.

Next, usefulness was also identified to mediate the relationship between social influence and students' attitudes toward MOOCs. The advantages of MOOCs should be recognised by the community [17] to help create social acceptance apart from influencing students' perceived usefulness and attitude toward MOOCs. The seventh effect is related to the full mediation of perceived ease of use in the relationship between the use of the Arabic language and perceived usefulness. A majority of MOOCs use English, making it difficult for those in the UAE whose local language is more dominant than English. However, the usefulness of MOOCs is not directly influenced by the use of local language, but also by the volubility of the courses that attract students' continuous use.

## 5. Implications

*5.1. Theoretical Contribution*

Despite the popularity of MOOCs and their future growth trend, many MOOCs that are provided by educational institutions suffer from sustainability issues because fewer than 10% of users completed their courses during the COVID-19 pandemic [4]. Such a low completion rate still forms a research gap, especially in developing countries where literature is scarce. Nevertheless, scholars from various disciplines attempted to bridge the gap of MOOCs' stability. Some of these studies emphasised the importance of behavioural factors [17,75], while others focused on technological and quality factors [76]. However, most previous of the literature focused more on the factors influencing users' intention to use MOOCs than on their continuous intention to use MOOCs. The factors influencing users' continuous intention to use MOOCs, which is the key driver for MOOCs' stability, still need to be investigated. Even though recent studies attempted to bridge the gap on the continuous intention to use MOOCs [17, 77–79], most of these studies addressed the continuance intention to use MOOCs in terms of behavioural factors or technological factors or both, without considering the contextual factors.

In this study, we argued that the integration between technological, social, and contextual factors could influence students' behaviour toward the continuous use of MOOCs.

Consequently, the TAM and TTF models that were applied to MOOCs in this study also incorporated the interaction with contextual factors. Not only does the current integrative model have more robust results than the TAM and TTF models do separately, but it also improves our awareness of TTF processes in contributing to MOOCs' development. In this regard, TAM combined with the TTF model, MOOC features, and social motivation could be regarded as a useful tool for studying actions in MOOCs.

The current study adds information to the established body of knowledge in four significant ways. This study was based on previous studies of MOOCs by emphasising the importance of achieving individual–technology fit and TTF. The findings of this study further indicated that perceived ease of use, usefulness, individual–technology fit, TTF, openness, reputation, social recognition, social influence, Arabic language use, and course advantage accreditation can all indirectly impact the continued intention to use MOOCs. The suggested integrated model offers a more comprehensive overview than the perspectives of individual TTF. This study, therefore, extends the body of knowledge of MOOCs' continuous intention through incorporating TAM and TTF theories.

This study also revealed that individual–technology fit, in addition to TTF, has a substantial impact on MOOCs' perceived usefulness. Therefore, this study can be used as a reference point for future studies on MOOCs using the TTF model. The proposed model also revealed a mechanism for determining the effects of TTF on MOOCs' perceived usefulness. Students are more likely to use MOOCs if they believe that it would benefit them in terms of TTF. Secondly, this study emphasised the value of using openness and reputation to manage perceived utility and ease of use. These elements can improve students' perceptions of usefulness and ease of use and, subsequently, their continued use of MOOCs. Of all the antecedent variables, reputation demonstrated the greatest effect on MOOCs' perceived usefulness. As a result, the relative significance of MOOCs in the proposed research model has been verified.

Next, this model suggested social motivation to improve MOOCs' perceived usefulness and attitude. Social motivation is reflected in social identity and group norms, where it involves social recognition and social impact. This study revealed that social recognition and social power can greatly impact perceived usefulness. As such, this study added to our knowledge about user cognition, in which previous studies primarily focused on technological adoption. According to the results, perceived utility is a key factor influencing students' attitudes toward the intention to use MOOCs in the future. Finally, MOOCs work differently from one context to another according to specific factors such as the use of language and course certificate advantages. However, most of the previous literature neglected these two factors, probably because a majority of the MOOCs' features deliver their courses in English with no certificate. In the UAE, however, using Arabic as an additional language facilitates the ease of using MOOCs, while course certificate advantage improves the usefulness of the users. Therefore, this study brings new insight to the body of knowledge by introducing new context factors. As a result, the element of use became critical in improving MOOC services to improve the long-term intention to use MOOCs. Hence, it is worth highlighting that the findings of this study contributed to a better understanding of the factors that influence students' intention to use MOOCs in the future.

### 5.2. Practical Contribution

According to the current findings, course openness with a low cost and social impact are the most important predictors for perceived usefulness in MOOCs. In terms of practice, several significant consequences and recommendations for MOOC practitioners can be suggested based on these results. Firstly, MOOC practitioners must understand that their decision to continue depends not just on their attitude toward MOOCs, but also on their perceived utility. Furthermore, perceived MOOC utility is a major mediator of the effects of perceived ease of use, task–technology suit, openness, and social motivation on continuation intention. Since perceived usefulness is the most important determinant of continuation intention, improving students' beliefs in the efficacy of MOOCs will increase

their intention to continue using them. In short, these results revealed that creating MOOCs with a modern interface and user-friendly screens would be insufficient to influence users' intention to continue. Instead, practitioners of MOOCs should emphasise practical functionality rather than the simplicity of use. Secondly, this study revealed that the TTF of MOOCs influenced perceived ease of use and usefulness, while individual–technology fit influenced perceived usefulness via perceived ease of use. Thus, MOOCs should clarify course prerequisites and challenges, such as the levels of required prior knowledge and the available resources to students.

Based on the analyses, practitioners of MOOCs should place more emphasis on ease of use and useful functionality. Secondly, this research shows that the TTF of MOOCs influences perceived ease of use and usefulness and that individual–technology fit influences perceived usefulness through perceived ease of use. As a result, MOOCs should be structured to explain course qualifications and obstacles, such as the levels of prior knowledge required and the resources available to students. Thirdly, openness and reputation are the two aspects MOOC providers can concentrate on to differentiate themselves from rivals (higher learning institutions) and increase an individual's perception of MOOCs by attracting students through continued use of MOOCs. Since most MOOCs are available at a low cost and there are a few switching costs, the impacts of cost efficiency on perceived usefulness are mediated by perceived ease of use. MOOC practitioners should concentrate on factors associated with the perceived ease of use by maximising accessible, rich multimedia capabilities to help promote a dynamic loop and share more user-generated experiences. Next, the perceived utility was also identified to be influenced by social motivation as a way to foster optimistic attitudes. MOOC professionals who apply these perspectives are more likely to attract and retain students in their courses. On the other hand, they will set themselves apart from the competition by ensuring that their courses are useful to learners. They must also value the effects of social influence, for which they may use peer influence to encourage continued use. With an increased sense of belonging to the company or community, one tends to adopt new technology. For instance, the intention to continue MOOCs can be strengthened through a fun, interactive learning atmosphere. Finally, since the use of the Arabic language in MOOCs can facilitate ease of use, the certificate advantages can influence users' usefulness perception, improving MOOCs' communication and accreditation and enhancing users' perception of MOOCs usefulness, which influences their continuous intention to use MOOCs.

*5.3. Limitations and Future Research*

Despite thorough and systematic analysis, a few drawbacks were identified in this study. This study was conducted in the UAE where MOOCs are still in their infancy. Therefore, self-selection bias could be present since the survey respondents voluntarily participated in this study. The ability to conduct random probabilistic sampling is only possible if the number of MOOC users grow. Secondly, this study employed a cross-sectional study. Hence, user behaviour is fluid. However, longitudinal evidence is also required to gain a better understanding of the inter-relationships or causality among the variables related to technology acceptance. Thirdly, determining causal effects among the constructs is difficult due to the cross-sectional nature of the study. Thus, future studies should add additional constructs such as TTF, MOOC functions, and social incentive constructs. As for the final drawback, the results and implications presented in this study were derived from a single study targeting a particular user community in the UAE, where the results were generalised for external validity. More studies will be needed to help generalise the observations by including various cultures where MOOCs are used [80].

## 6. Conclusions

MOOCs have been recently developed as an educational technology system to provide online courses providing unrestricted access to users and participation in any course

of their choice. Moreover, MOOCs provide a venue for interactive forums compared to the traditional modalities of teaching such as lectures, videos, and reading materials. However, continuance intention to use MOOCs still forms a challenge to both MOOC providers and developers, as only 10% of the enrolled students finished their courses. Even though MOOCs studies have been recently growing, leaving a wide literature, most of the past studies paid attention to the effect of either technological factors or motivational factors on students' continuance intention to use. In addition, most of the past studies were conducted in European countries, and only a few Asian countries. MOOCs literature in developing countries, especially in the Arab region, is still limited. This leaves a gap since MOOCs work differently from one country to another due to the influence of contextual factors, which were not well-discussed in the literature. Thus, driven by TAM and TTF theories, this study aims to bridge these gaps through examining the relationship between technological and social motivational factors as well as MOOCs' contextual features and students' perceived ease of use and usefulness, which explain students' attitude toward MOOCs and continuance intention to use. Driven by a survey gathered from 315 of the UAE's students, this study revealed that the technological factors of task-fit technology as well as individual-fit technology are important indicators toward students' continuance intention to use MOOCs because they influence students' perceived ease of use and usefulness. Second, the findings of this study revealed that the social motivational factors driven by social influence and recognition are important indicators toward students' continuance intention to use MOOCs because they influence their behaviour and attitude. Finally, this study showed that MOOCs' contextual features, such as MOOCs openness, reputation, Arab language use, and course certificate accreditation, are important indicators that explain students' continuance intention to use MOOCs through influencing their perception on MOOCs' ease of use and usefulness.

**Author Contributions:** Conceptualization and formal analysis, Writing—original draft, S.A.A.-s.; Methodology and data curation, S.A.; Investigation, M.K.; Resources and Data curation, N.H.A.-K.; Validation and analysis, A.A.M.; Writing—review & editing, M.A.-s. Funding acquisition and Visualization, M.M.J. All authors have read and agreed to the published version of the manuscript.

**Funding:** This research received no external funding.

**Institutional Review Board Statement:** Local ethics committees (Universiti Teknikal Malaysia Melaka) ruled that no formal ethics approval was required in this particular case. This study has been performed in accordance with the Declaration of Helsinki.

**Informed Consent Statement:** Informed consent for participation was obtained from respondents who participated in the survey. The respondents who participated in the survey were asked to read the ethical statement ("There is no compensation for responding nor is there any known risk. To ensure that all information will remain confidential, please do not include your name. Participation is strictly voluntary and you may refuse to participate at any time") and proceed only if they agree. No data were collected from anyone under 16 years old.

**Data Availability Statement:** Data are available upon request from researchers who meet the eligibility criteria. Kindly contact the first author privately through e-mail.

**Acknowledgments:** we would like to thank Universiti Tekinkal Malaysia Melaka for technical supporting

**Conflicts of Interest:** The authors declare no conflict of interest.

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
