# Peer review of "A Model of Motivational and Technological Factors Influencing Massive Open Online Courses’ Continuous Intention to Use"

_sustainability, doi:10.3390/su14159279_

Round 1
Reviewer 1 Report
1) The abstract should be rewritten. The abstract should be written as a continuous paragraph with 120-150 words and recapitulative state the background of the research, purpose, methodologies, major conclusions, and its contributions to the field. It should emphasize new or important aspects of the study. Do not use any statistical sign or number. In addition no citations and references…
2) What is the purpose of this research? The introduction must be written at the end of the section.
3) The argument part is very poorly written. The results of other research should be compared by comparison.
4) Table 4 is very long. The table should be divided into 4 sections and explained. A spreadsheet should be at least one page long.
Author Response
Thank you very much for your valuable comments. We confirm that we have addressed all comments and we hope the new version will meet your satisfaction. Please see attached file, which provides a further explanation about the comments

Reviewer 2 Report
The article is interesting but it would be more effective if it did a study in connection with MOOCs and their characteristics.
Some comments should indicate the source of the arguments, for example, "Massive Open Online Courses (MOOCs) are an EFFECTIVE innovative technology..."
The text should have a simpler wording that clearly shows the main idea of ​​each paragraph. Reading becomes difficult. There are also many repetitions of terms, for example, "user", "factor", "social", etc.etc.
Perhaps it should be explained that UAE stands for United Arab Emirates.
Some references have no connection to MOOCs, for example Reference 10 is old and deals with scales for predicting user acceptance of computers.
It would be interesting to have access to the original data of the instruments/questionnaire. This would help to understand the DATA ANALYSIS chapter, which is a statistical chapter.
It is not understood why it starts with a numbering in 4.3. Path analysis; 4.4. Mediating effects 5. Discussion
Repetitive writing, limited contributions to the scientific and academic community.
Author Response
Thank you for your valuable comments. We confirmed that we have addressed your comments as shown in the attached report along with the new version, which we hope that it will meet your satisfactory

Reviewer 3 Report
1. Abstract: The summary should follow the style of structured summaries (background, methods, results, and conclusions but no heading).
2. The general objective and specific objectives should appear at the end of the introduction. The objective should be clearly written, referring to the population, the intervention, the comparison and the results
3. Methodology/ Did you realize a sample size estimation? If not, why?
4. Discussion/Although the main finding has been correctly reported at the beginning of this section, a short remarking on the aim of the study could be reported just to provide more impact to the results of the study. I recommend adding one or two more specific conclusion to highlight your main results of study.
5. Discussion: the whole discussion section should be improved and expanded. It mainly reports other articles’ results, while the actual findings are poorly discussed and compared with the present literature. Furthermore, the results are not critically discussed, and no practical applications are proposed.
Author Response

(The authors gave the same response as above.)

Round 2
Reviewer 1 Report
1) Do not use abbreviations in the title and abstract.
2) Do not use monthly keywords more than once in the abstract. For example MOOC.....
3) Do not use City, Country, or University names in the title, abstract, and text. You can only use it under Participants in Method. Advertising has no place in science.
4) Do not write the tables one after the other. After writing a table, go to the explanation path. After explaining that table, including the other table.
What you give creates a problem for the reader to follow.
You can benefit from the results of the following articles. They are related to MOOC.
1) https://un-pub.eu/ojs/index.php/wjet/article/view/4507
2) https://doi.org/10.18844/wjet.v8i3.691
Author Response
Dear reviewer,
Greetings and thank you for your time and effort in reviewing our manuscript. We would like to inform you we have improved our article as shown in the attached file.

Reviewer 2 Report
I appreciate the acceptance of the proposals.
I think your point of view and context UAE of the research you present is appropriate and necessary.
Congratulations!
Author Response
Dear reviewer,
Greetings and thank you for your effort in reviewing our manuscript. Thank you also for your recommendation.

Reviewer 3 Report
Authors had answers to the comments and clarified the reviewer’s questions. this study proposed an integrative model to explain ways to improve students’ continuance usage intention through integrative technological, I think this study is sound and should be published.
Author Response
Dear reviewer,
Greeting and wish you a wonderful time,
Thank you for your time in reviewing our manuscript and your kind recommendation. We have improved the manuscript as shown in the new verision
